# Q-DM: An Efficient Low-bit Quantized Diffusion Model

**Yanjing Li[1†‡], Sheng Xu[1†], Xianbin Cao[1∗], Xiao Sun[2∗], Baochang Zhang[1,3,4]**
[1]Beihang University
[2]Shanghai Artificial Intelligence Laboratory
[3]Zhongguancun Laboratory
[4] Nanchang Institute of Technology
{yanjingli, shengxu}@buaa.edu.cn

## Abstract

Denoising diffusion generative models are capable of generating high-quality data, but suffers from the computation-costly generation process, due to a iterative noise estimation using full-precision networks. As an intuitive solution, quantization can significantly reduce the computational and memory consumption by low-bit parameters and operations. However, low-bit noise estimation networks in diffusion models (DMs) remain unexplored yet and perform much worse than the full-precision counterparts as observed in our experimental studies. In this paper, we first identify that the bottlenecks of low-bit quantized DMs come from a large distribution oscillation on activations and accumulated quantization error caused by the multi-step denoising process. To address these issues, we first develop a Timestep-aware Quantization (TaQ) method and a Noise-estimating Mimicking (NeM) scheme for low-bit quantized DMs (Q-DM) to effectively eliminate such oscillation and accumulated error respectively, leading to well-performed low-bit DMs. In this way, we propose an efficient Q-DM to calculate low-bit DMs by considering both training and inference process in the same framework. We evaluate our methods on popular DDPM and DDIM models. Extensive experimental results show that our method achieves a much better performance than the prior arts. For example, the 4-bit Q-DM theoretically accelerates the 1000-step DDPM by $7.8\times$ and achieves a FID score of 5.17, on the unconditional CIFAR-10 dataset.

## 1 Introduction

Denoising diffusion models, also known as score-based generative models [10, 33, 35], have recently shown remarkable success in various generative tasks such as images [10, 35, 22], audio [21], video [31], and graphs [23]. These models have also demonstrated flexibility in downstream tasks, making them attractive for tasks such as super-resolution [26, 7] and image-to-image translation [29]. Compared to Generative Adversarial Networks (GANs) [8], historically considered state-of-the-art, diffusion models have proven to be superior in terms of quality and diversity in most of these tasks and applications. The process of diffusion models involves gradually transforming real data into Gaussian noise, which is then reversed via a denoising process to generate real data [10, 40]. However, such denoising process is time-consuming and involves iterating a neural network for noise estimation over thousands of timesteps, despite producing a significant amount of images. Therefore, researchers are actively working on accelerating this generation process to reduce its long iterative process and high inference cost for sample generation. To achieve this, one pipeline is to

---

† Equal contribution. ∗ Corresponding author.
‡ This work was done during her internship at Shanghai Artificial Intelligence Laboratory.

37th Conference on Neural Information Processing Systems (NeurIPS 2023).

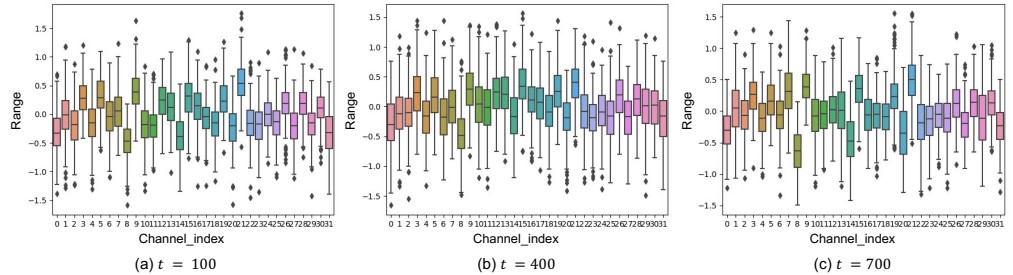

Figure 1: Studies on the activation distribution *w.r.t.* time-step. Per (output) channel activation ranges of the first attention block in diffusion model on different timestep. The boxplot visualizes key statistical measures for each channel, including the minimum and maximum values, the 2nd and 3rd quartiles, and the median.

focus on sample trajectory learning, to develop faster sampling strategies [28, 22, 1]. While the other pipeline directly compresses and accelerates the noise estimation networks based on network quantization technology [30], which is particularly suitable for AI chips because of the low-bit parameters and operations. Prior post-training quantization (PTQ) methods [30, 19, 17] on diffusion models (DMs) or other neural networks directly compute quantized parameters based on pre-trained full-precision models, which constrains the model performance to a sub-optimized level without fine-tuning. Furthermore, quantizing DMs based on PTQ methods to ultra-low bits (*e.g.*, 4 bits or lower) is ineffective and suffers from a significant performance reduction.

Differently, quantization-aware training (QAT) [16, 18] methods perform quantization during back propagation and generally achieve a less performance drop with a higher compression rate than PTQ. For instance, QAT has been shown to be effective for CNNs [5, 18] and ViTs [16, 18] and BERT [24]. However, QAT methods for low-bit quantization of diffusion models remain largely unexplored. Therefore, we first build a low-bit quantized DM baseline, a straightforward yet effective solution based on common techniques [5]. Our experimental studies reveal that the severe performance drop of low-bit quantized DMs, such as PTQ [30] and baseline [5], lies in the activation distribution oscillation and quantization error accumulation caused by the denoising process.

As shown in Fig. 1, the output distribution of the noise estimation network at each time step can differ significantly, resulting in activation distribution oscillation. Particularly, the distribution of activation in a specific layer varies significantly across different timesteps during training. We also observe that errors between full-precision activations and quantized activations gradually accumulate across timesteps during the sampling process (inference), making it harder to produce well-performed quantized DMs.

Drawing on the aforementioned insights, we propose a Timestep-aware Quantization (TaQ) method to address the oscillating distribution issue. By smoothing out these fluctuations and introducing more precise scaling factors into activations, we effectively enhance the performance of the low-bit quantized DMs. We further design a new training scheme for quantized DMs, dubbed Noise-estimating Mimicking (NeM), which can reduce the accumulated errors and promote the performance of quantized DMs based on the knowledge of full-precision counterparts. In this way, we achieve a new QAT method for low-bit quantized DM (Q-DM) via incorporating all the explorations (see the overview in Fig. 2). Overall, the contributions of this paper can be summarized as follows:

- To the best of our knowledge, we proposed the first QAT method towards efficient low-bit DMs, dubbed Q-DM, by fully considering both training and inference process in the same framework.

- We introduce a Timestep-aware Quantization (TaQ) method to mitigate activation distribution oscillation caused by the random-sampled timestep in the training process. We develop a Noise-estimating Mimicking (NeM) scheme to reduce accumulated errors, by which the Q-DMs are able to achieve comparable performance as the full-precision counterparts.

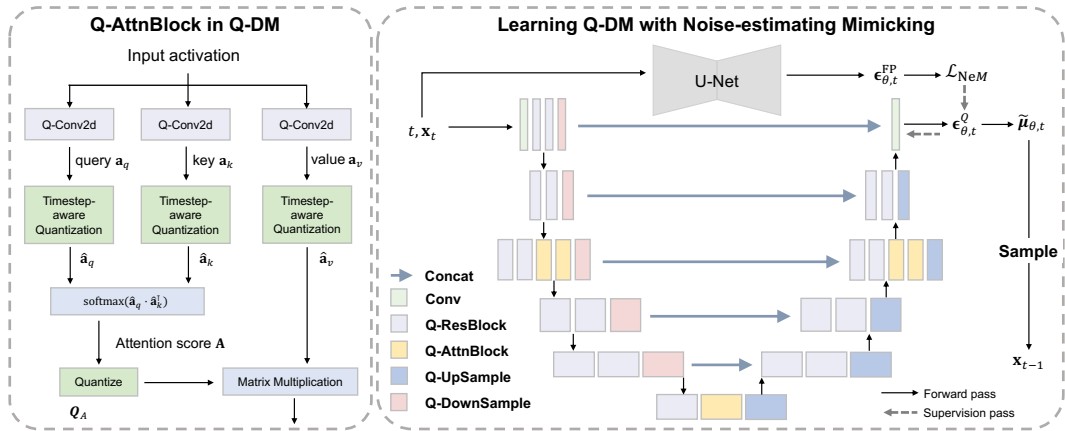

Figure 2: Overview of the proposed Q-DM framework. We introduce the timestep-aware quantization in an architecture perspective and a noise imitation training scheme incorporated in the optimization process. From left to right, we respectively show the detailed architecture of single Q-AttnBlock in Q-DM and the training framework of Q-DM.

- Extensive experiments on the CIFAR-10 and ImageNet datasets show that our Q-DM outperforms the baseline and 8-bit PTQ method by a large margin, and achieves comparable performances as the full-precision counterparts with a considerable acceleration rate.

## 2 Related Work

**Network Quantization**. Quantizing neural networks (QNNs) often possess low-bit ($1 \sim 4$-bit) weights and activations to accelerate the model inference and save the memory usage. Specifically, ternary weights are introduced to reduce the quantization error in TWN [15]. DoReFa-Net [41] exploits convolution kernels with low bit-width parameters and gradients to accelerate both the training and inference. TTQ [42] uses two full-precision scaling coefficients to quantize the weights to ternary values. Zhuang *et al.* [43] present a $2 \sim 4$-bit quantization scheme using a two-stage approach to alternately quantize the weights and activations, which provides an optimal trade-off among memory, efficiency, and performance. Jung *et al.* [12] parameterize the quantization intervals and obtain their optimal values by directly minimizing the task loss of the network and also the accuracy degeneration with further bit-width reduction. ZeroQ [2] supports both uniform and mixed-precision quantization by optimizing for a distilled dataset, which is engineered to match the statistics of batch normalization across different layers of the network. Xie *et al.* [39] introduces transfer learning into network quantization to obtain an accurate low-precision model by utilizing the Kullback-Leibler (KL) divergence. PWLQ [6] enables accurate approximation for tensor values that have bell-shaped distributions with long tails and finds the entire range by minimizing the quantization error.

**Diffusion Model**. The high cost of denoising through networks and the long iterative process make it difficult to implement diffusion models widely. To accelerate diffusion probabilistic models (DMs) [10], previous research has focused on finding shorter sampling trajectories while maintaining DM performance. Wavegrad [3] introduces grid search, which finds an effective trajectory with only six timesteps, but this approach cannot be generalized for longer trajectories due to its exponentially growing time complexity. Watson *et al.* [38] model the trajectory searching as a dynamic programming problem. Song *et al.* [34] construct non-Markovian diffusion processes that lead to the same training objective, but whose reverse process can be much faster to sample from. For DMs with continuous timesteps, Song *et al.* [33, 35] have formulated the DM in the form of an ordinary differential equation (ODE) and improved sampling efficiency by using faster ODE solvers. Jolicoeur-Martineau *et al.* [11] have introduced an advanced SDE solver to accelerate the reverse process via an adaptively larger sampling rate. Analytic-dpm [1] has estimated variance and KL divergence using the Monte Carlo method and a pretrained score-based model with derived analytic forms that are simplified from the score-function. In addition to those training-free methods, Luhman & Luhman [20] have

compressed the reverse denoising process into a single-step model, while San-Roman *et al.* [28] has dynamically adjusted the trajectory during inference. However, implementing these methods requires additional training after obtaining a pretrained DM, which makes them less desirable in most situations. In summary, all these DM acceleration methods can be categorized as finding effective sampling trajectories.

Unlike prior works, we demonstrate that diffusion models can be accelerated by compressing the network in each noise estimating iteration, which is orthogonal with the fast sampling methods mentioned above. To the best of our knowledge, this is the first study to explore low-bit quantized diffusion models in a quantization-aware training (QAT) manner.

## 3 Background and Challenge

### 3.1 Diffusion Models

**Forward process**. Let $\mathbf{x}_0$ be a sample from the data distribution $\mathbf{x}_0 \sim q(\mathbf{x})$. A forward diffusion process adds Gaussian noise to the sample for $T$ times, resulting in a sequence of noisy samples $\mathbf{x}_1, \cdots, \mathbf{x}_T$ as:

$$q(\mathbf{x}_t|\mathbf{x}_{t-1}) = \mathcal{N}(\mathbf{x}_t; \sqrt{1-\beta_t}\mathbf{x}_{t-1}, \beta_T\mathbf{I}), \tag{1}$$

where $\beta_t \in (0,1)$ is the variance schedule and controls the strength of the Gaussian noise in each step. The forward diffusion process satisfies the Markov property since each step relies solely on the preceding step. Additionally, as the number of steps increases towards infinity ($T \to \infty$), the final state $\mathbf{x}_T$ converges to an isotropic Gaussian distribution. A notable property of the forward process is that it admits sampling $\mathbf{x}_t$ at an arbitrary timestep $t$ in closed form as:

$$q(\mathbf{x}_t|\mathbf{x}_0) = \mathcal{N}(\mathbf{x}_t; \sqrt{\bar{\alpha}_t}\mathbf{x}_0, (1-\bar{\alpha}_t)\mathbf{I}). \tag{2}$$

**Reverse process**. To generate a sample from a Gaussian noise input $\mathbf{x}_T \sim \mathcal{N}(\mathbf{0}, \mathbf{I})$ using diffusion models, the forward process is reversed. However, since the actual reverse conditional distribution $q(\mathbf{x}_{t-1}|\mathbf{x}_t)$ is unknown, diffusion models use a learned conditional distribution $p_\theta(\mathbf{x}_{t-1}|\mathbf{x}_t)$ that approximates the real reverse conditional distribution with a Gaussian distribution. This approximation is expressed as:

$$p_\theta(\mathbf{x}_{t-1}|\mathbf{x}_t) = \mathcal{N}(\mathbf{x}_{t-1}; \tilde{\boldsymbol{\mu}}_{\theta,t}(\mathbf{x}_t), \tilde{\beta}_t\mathbf{I}). \tag{3}$$

By using the re-parameterization trick presented in [10], it becomes possible to derive the mean $\tilde{\boldsymbol{\mu}}_{\theta,t}(\mathbf{x}_t)$ and $\tilde{\beta}_t\mathbf{I}$ as follows:

$$\begin{aligned} \tilde{\boldsymbol{\mu}}_{\theta,t}(\mathbf{x}_t) &= \frac{1}{\sqrt{\alpha_t}}(\mathbf{x}_t - \frac{1-\alpha_t}{\sqrt{1-\bar{\alpha}_t}}\boldsymbol{\epsilon}_{\theta,t}), \\ \tilde{\beta}_t &= \frac{1-\bar{\alpha}_{t-1}}{1-\bar{\alpha}_t} \cdot \beta_t, \end{aligned} \tag{4}$$

where $\alpha_t = 1 - \beta_t, \bar{\alpha}_t = \prod_{i=1}^{t} \alpha_i$ and $\boldsymbol{\epsilon}_\theta$ is a function approximator intended to predict $\boldsymbol{\epsilon}$ from $\mathbf{x}_t$ [10].

**Training**. At training time, the goal of optimization is to minimize the negative log-likelihood, *i.e.*, $-\log p_\theta(\mathbf{x}_0)$. With variational inference, a lower bound of it could be found, denoted as $L_{\text{VLB}}$:

$$L_{\text{VLB}} = \mathbb{E}_{q(\mathbf{x}_{0:T})}[\log \frac{q(\mathbf{x}_{1:T}|\mathbf{x}_0)}{p_\theta(\mathbf{x}_{0:T})}] \geq -\log p_\theta(\mathbf{x}_0). \tag{5}$$

It is found in [10] that using a simplified loss function to $L_{\text{VLB}}$ often obtains better performance:

$$L_{\text{simple}} = \mathbb{E}_{t,\mathbf{x}_0,\boldsymbol{\epsilon}_t}[\|\boldsymbol{\epsilon}_t - \boldsymbol{\epsilon}_\theta(\sqrt{\bar{\alpha}_t}\mathbf{x}_0 + \sqrt{1-\bar{\alpha}_t}\boldsymbol{\epsilon}_t, t)\|^2]. \tag{6}$$

**Sampling**. At inference time, a Gaussian noise tensor $\mathbf{x}_T$ is sampled and is denoised by repeatedly sampling the reverse distribution $p_\theta(\mathbf{x}_{t-1}|\mathbf{x}_t)$. $\tilde{\boldsymbol{\mu}}_{\theta,1}(\mathbf{x}_1)$ is taken as the final generation result, with no noise added in the final denoising step.

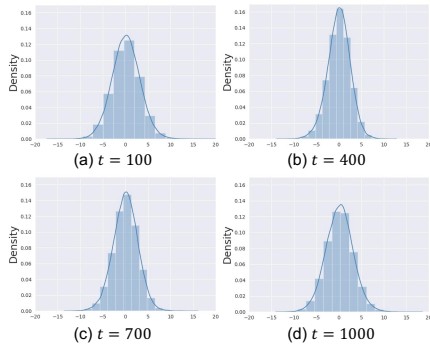
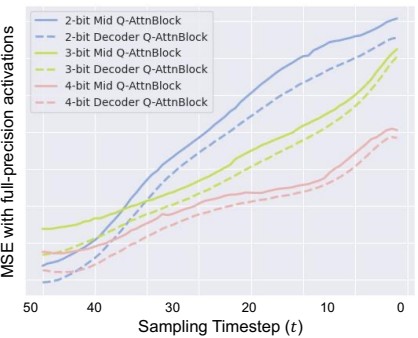

Figure 3: Input activation distribution of the first Q-AttnBlock in diffusion model on different timestep with a model trained on CIFAR-10 [13] by DDPM.

Figure 4: Distance between the outputs of the full-precision model and different bit-width baseline models trained on CIFAR-10 [13] by DDIM with 100 sampling steps.

## 3.2 Quantization

Given an $N$-layer CNN model, we denote its weight set as $\mathbf{W} = \{\mathbf{w}^n\}_{n=1}^N$ and input feature map set as $\mathbf{A} = \{\mathbf{a}_{\text{in}}^n\}_{n=1}^N$. The $\mathbf{w}^n \in \mathbb{R}^{C_{\text{out}}^n \times C_{\text{in}}^n \times K^n \times K^n}$ and $\mathbf{a}_{\text{in}}^n \in \mathbb{R}^{C_{\text{in}}^n \times W_{\text{in}}^n \times H_{\text{in}}^n}$ are the convolutional weight and the input feature map in the $n$-th layer, where $C_{\text{in}}^n$, $C_{\text{out}}^n$ and $K^n$ respectively stand for input channel number, output channel number and the kernel size. Also, $W_{\text{in}}^n$ and $H_{\text{in}}^n$ are the width and height of the feature maps. Then, the convolutional outputs $\mathbf{a}_{\text{out}}^n$ can be technically formulated as:

$$\mathbf{a}_{\text{out}}^n = \mathbf{w}^n \otimes \mathbf{a}_{\text{in}}^n, \tag{7}$$

where $\otimes$ represents the convolution operation. Herein, we omit the non-linear function for simplicity. Quantized neural network intends to represent $\mathbf{w}^n$ and $\mathbf{a}^n$ in a low-bit format such that the float-point convolutional outputs can be approximated as:

$$\hat{\mathbf{w}}^n = \boldsymbol{s}^{w^n} \circ Q(\mathbf{w}^n) = \boldsymbol{s}^{w^n} \circ \lfloor \text{clip}(\mathbf{w}^n/\boldsymbol{s}^{w^n}, -2^{b-1}, 2^{b-1} - 1) \rceil$$
$$\hat{\mathbf{a}}_{\text{in}}^n = s^{a_{\text{in}}^n} \cdot Q(\mathbf{a}_{\text{in}}^n) = s^{a_{\text{in}}^n} \cdot \lfloor \text{clip}(\mathbf{a}_{\text{in}}^n/s^{a_{\text{in}}^n}, -2^{b-1}, 2^{b-1} - 1) \rceil \tag{8}$$
$$\mathbf{a}_{\text{out}}^n = \hat{\mathbf{a}}_{\text{in}}^n \otimes \hat{\mathbf{w}}^n \approx s^{a_{\text{in}}^n} \cdot \boldsymbol{s}^{w^n} \circ [Q(\mathbf{w}^n) \odot Q(\mathbf{a}_{\text{in}}^n)],$$

where $\circ$ denotes the channel-wise multiplication, $\odot$ denotes the efficient GEMM operations, and $\boldsymbol{s}^{w^n} = \{\boldsymbol{s}_1^{w^n}, \boldsymbol{s}_2^{w^n}, ..., \boldsymbol{s}_{C_{\text{out}}^n}^{w^n}\} \in \mathbb{R}_+^{C_{\text{out}}^n}$ is known as the channel-wise scaling factor vector [25] to mitigate the output gap between Eq. (7) and its approximation of Eq. (8). Meanwhile, we use the layer-wise quantization for input activations and the scaling factor of activations $s^{a_{\text{in}}^n} \in \mathbb{R}_+$ is a scalar.

## 3.3 Challenge Analysis

Here we identify two major challenges on low-bit DMs, specific to the multi-step inference process and random-sampled-step training process of diffusion models. Namely, we investigate on the distribution oscillation of the activations, and the accumulated quantization error resulted from the multi-step denoising process.

**Activation distribution oscillation**. To understand the distribution change of diffusion models, we investigate the activation distribution, *w.r.t.* timestep in the training process. Theoretically, if the distribution changes *w.r.t.* timestep, it would be difficult to implement previous QAT methods. We analyze the overall activation distributions of the noise estimation network, as shown in Fig. 3. We can observe that at different timesteps, the corresponding activation distributions have large discrepancies, *e.g.*, Fig. 3(a) *v.s.* Fig. 3(b), which makes previous QAT methods [16] in-applicable for multi-timestep models, *i.e.*, diffusion models.

**Quantization error accumulation**. Quantization of a noise estimation network introduces disturbances to the weights and activations, resulting in errors in each layer's output. Previous studies [4]

have found that these errors tend to accumulate across layers, making it more challenging to quantize deeper neural networks. In the case of diffusion models (DMs), at each time step $t$, the input of the model ($\mathbf{x}_{t-1}$) is obtained from the model's output at the previous time step $t$, *i.e.*, $\mathbf{x}_t$. As depicted in Fig. 4, the MSE distance, representing the quantization error of low-bit quantized DMs, exhibits a noticeable growth along with the decrease of sampling timestep. This implies that as the denoising process moves towards later timestep, the accumulation of quantization errors becomes more prominent.

## 4 The Proposed Q-DM

### 4.1 Timestep-aware Quantization

To tackle the distribution oscillation in the training process, we first introduce the quantized attention block that efficiently takes into account the timestep. This structure allows for the numerical analysis of activation ranges across different timesteps and mitigates distribution oscillation of low-bit quantized DMs. We recall the quantization in the attention block based on Eq. (8), which is formulated as:

$$
\begin{aligned}
&\hat{\mathbf{a}}_q(\mathbf{x}_t, t) = s^{a_q(\mathbf{x}_t, t)} \cdot Q(\mathbf{a}_q(\mathbf{x}_t, t)), \ \ \hat{\mathbf{a}}_k(\mathbf{x}_t, t) = s^{a_k(\mathbf{x}_t, t)} \cdot Q(\mathbf{a}_k(\mathbf{x}_t, t)) \\
&\mathbf{A}(\mathbf{x}_t, t) = \mathrm{softmax}[(\hat{\mathbf{a}}_q(\mathbf{x}_t, t) \cdot \hat{\mathbf{a}}_k(\mathbf{x}_t, t)^\top)/\sqrt{d}], \\
&\hat{\mathbf{A}}(\mathbf{x}_t, t) = s^{A(\mathbf{x}_t, t)} \cdot Q(\mathbf{A}(\mathbf{x}_t, t)), \\
&\mathbf{a}_{\mathrm{out}}(\mathbf{x}_t, t) = \hat{\mathbf{A}}(\mathbf{x}_t, t) \cdot \hat{\mathbf{a}}_v(\mathbf{x}_t, t)^\top,
\end{aligned}
\tag{9}
$$

where $\mathbf{A}$ is the attention score.

In the $i$-th mini-batch, the timestep is represented as $\{t_1, \cdots, t_{b_i}\}$, where $b_i$ is the batch size of the $i$-th batch. We denote $i \in \{1, \cdots, B\}$, and $B$ is the number of batch. Therefore, we calculate the timestep-aware distribution divergence for the query activation $\mathbf{a}_q$ as:

$$
\begin{aligned}
\gamma_{q;t} &= \sum_{i=1}^{B} \frac{1}{b_i} \sum_{j=1}^{b_i} \mathbf{a}_q(\mathbf{x}_{t_j}, t_j), \\
\sigma_{q;t}^2 &= \sum_{i=1}^{B} \frac{1}{b_i} \sum_{j=1}^{b_i} [\mathbf{a}_q(\mathbf{x}_{t_j}, t_j) - \gamma_{q;t}]^2,
\end{aligned}
\tag{10}
$$

where $\gamma_{q;t}$ and $\sigma_{q;t}^2$ are statistical mean and variance of query activation $\mathbf{a}_q$. And the calculation of the key activation $\mathbf{a}_k$ is likewise.

Based on such statistical results, the query and key activations in each specific timestep are smoothed as:

$$
\begin{aligned}
\tilde{\mathbf{a}}_q(\mathbf{x}_t, t) &= [\mathbf{a}_q(\mathbf{x}_t, t) - \gamma_{q;t}]/\sqrt{\sigma_{q;t}^2 + \psi} \\
\tilde{\mathbf{a}}_k(\mathbf{x}_t, t) &= [\mathbf{a}_k(\mathbf{x}_t, t) - \gamma_{k;t}]/\sqrt{\sigma_{k;t}^2 + \psi},
\end{aligned}
\tag{11}
$$

where $\psi$ is constant to avoid 0 denominator. With the above timestep-aware smoothing process, we formulate our timestep-aware quantization as:

$$
\begin{aligned}
&\hat{\mathbf{a}}_q(\mathbf{x}_t, t) = s^{a_q(\mathbf{x}_t, t)} \cdot \mathrm{TaQ}(\mathbf{a}_q(\mathbf{x}_t, t)), \ \ \hat{\mathbf{a}}_k(\mathbf{x}_t, t) = s^{a_k(\mathbf{x}_t, t)} \cdot \mathrm{TaQ}(\mathbf{a}_k(\mathbf{x}_t, t)) \\
&\mathbf{A}(\mathbf{x}_t, t) = \mathrm{softmax}[(\hat{\mathbf{a}}_q(\mathbf{x}_t, t) \cdot \hat{\mathbf{a}}_k(\mathbf{x}_t, t)^\top)/\sqrt{d}], \\
&\hat{\mathbf{A}}(\mathbf{x}_t, t) = s^{A(\mathbf{x}_t, t)} \cdot \mathrm{TaQ}(\mathbf{A}(\mathbf{x}_t, t)), \\
&\mathbf{a}_{\mathrm{out}}(\mathbf{x}_t, t) = \hat{\mathbf{A}}(\mathbf{x}_t, t) \cdot \hat{\mathbf{a}}_v(\mathbf{x}_t, t)^\top,
\end{aligned}
\tag{12}
$$

in which $\mathrm{TaQ}(*) = \lfloor \mathrm{clip}([* - \gamma_{*;t}]/[s^* \cdot \sqrt{\sigma_{*;t}^2 + \psi}], -2^{b-1}, 2^{b-1} - 1) \rceil$. The smoothed activations are less sensitive to the random sampled timestep in the trianing process and the timestep-aware quantization, to some extent, dismisses the distribution oscillation phenomenon.

Table 1: Evaluating the components of Q-DM based on 50-step DDIM sampler with $32\times32$ generating resolution on CIFAR-10 [13]. "#Bits" denotes bit-width of weights and activations

| Method | #Bits | FID↓ | IS↑ | #Bits | FID↓ | IS↑ | #Bits | FID↓ | IS↑ |
|--------|-------|------|-----|-------|------|-----|-------|------|-----|
| Full-precision | 32-32 | 4.67 | 9.27 | - | - | - | - | - | - |
| PTQ4DM | 8-8 | 18.02 | 8.87 | - | - | - | - | - | - |
| Baseline (LSQ [5]) | 4-4 | 10.22 | 8.91 | 3-3 | 13.24 | 8.88 | 2-2 | 18.74 | 8.65 |
| +TaQ | 4-4 | 9.25 | 8.95 | 3-3 | 11.19 | 8.91 | 2-2 | 16.83 | 8.71 |
| +NeM | 4-4 | 8.98 | 8.92 | 3-3 | 11.02 | 8.90 | 2-2 | 16.97 | 8.79 |
| **+TaQ+NeM (Q-DM)** | 4-4 | **6.89** | **8.96** | 3-3 | **9.07** | **8.98** | 2-2 | **15.26** | **8.86** |

## 4.2 Noise-estimating Mimicking

To mitigate the negative impact of quantization error accumulation on the training of a quantized DM $\theta^{\mathrm{Q}}$, a full-precision DM, denoted as $\theta^{\mathrm{FP}}$, is incorporated into the training process to facilitate the learning objective. Following [10], with $p_{\theta^{\mathrm{Q}}}(\mathbf{x}_{t-1}|\mathbf{x}_t) = \mathcal{N}(\mathbf{x}_{t-1}; \tilde{\boldsymbol{\mu}}_{\theta^{\mathrm{Q}},t}(\mathbf{x}_t), \tilde{\beta}_t\mathbf{I})$ and $p_{\theta^{\mathrm{FP}}}(\mathbf{x}_{t-1}|\mathbf{x}_t) = \mathcal{N}(\mathbf{x}_{t-1}; \tilde{\boldsymbol{\mu}}_{\theta^{\mathrm{FP}},t}(\mathbf{x}_t), \tilde{\beta}_t\mathbf{I})$, we can write:

$$L_{t-1} = \mathbb{E}_q\left[\frac{1}{2\tilde{\beta}_t}\|\boldsymbol{\mu}_{\theta^{\mathrm{FP}}}(\mathbf{x}_t, t) - \boldsymbol{\mu}_{\theta^{\mathrm{Q}}}(\mathbf{x}_t, t)\|^2\right] + C, \tag{13}$$

where $C$ is a constant that does not depend on $\theta^{\mathrm{Q}}$ or $\theta^{\mathrm{FP}}$. As in Eq. (13), we aim to compel the quantized model to replicate the noise estimation capability of the full-precision model. Further, by re-parameterizing Eq. (2) as $\mathbf{x}_t(\mathbf{x}_0, \boldsymbol{\epsilon}) = \sqrt{\bar{\alpha}_t}\mathbf{x}_0 + \sqrt{1-\bar{\alpha}_t}\boldsymbol{\epsilon}$ for $\boldsymbol{\epsilon} \sim \mathcal{N}(\mathbf{0}, \mathbf{I})$ and following the formulation in [10], which utilizes the formula for the posterior of the forward process, we can derive that:

$$L_{t-1} - C = \mathbb{E}_{\mathbf{x}_0, \boldsymbol{\epsilon}}\left[\frac{1}{2\tilde{\beta}_t}\|\boldsymbol{\mu}_{\theta^{\mathrm{FP}}}(\mathbf{x}_t(\mathbf{x}_0, \boldsymbol{\epsilon}), t) - \boldsymbol{\mu}_{\theta^{\mathrm{Q}}}(\mathbf{x}_t(\mathbf{x}_0, \boldsymbol{\epsilon}), t)\|^2\right], \tag{14}$$

where $\boldsymbol{\mu}_{\theta^{\mathrm{FP}}}(\mathbf{x}_t(\mathbf{x}_0, \boldsymbol{\epsilon}), t)$ and $\boldsymbol{\mu}_{\theta^{\mathrm{Q}}}(\mathbf{x}_t(\mathbf{x}_0, \boldsymbol{\epsilon}), t)$ are parameterized as:

$$\begin{aligned}
\boldsymbol{\mu}_{\theta^{\mathrm{FP}}}(\mathbf{x}_t, t) &= \tilde{\boldsymbol{\mu}}_t\left(\mathbf{x}_t, \frac{1}{\sqrt{\bar{\alpha}_t}}[\mathbf{x}_t - \sqrt{1-\bar{\alpha}_t}\boldsymbol{\epsilon}_{\theta^{\mathrm{FP}}}(\mathbf{x}_t)]\right) = \frac{1}{\alpha_t}\left(\mathbf{x}_t - \frac{\beta_t}{\sqrt{1-\bar{\alpha}_t}}\boldsymbol{\epsilon}_{\theta^{\mathrm{FP}}}(\mathbf{x}_t, t)\right), \\
\boldsymbol{\mu}_{\theta^{\mathrm{Q}}}(\mathbf{x}_t, t) &= \tilde{\boldsymbol{\mu}}_t\left(\mathbf{x}_t, \frac{1}{\sqrt{\bar{\alpha}_t}}[\mathbf{x}_t - \sqrt{1-\bar{\alpha}_t}\boldsymbol{\epsilon}_{\theta^{\mathrm{Q}}}(\mathbf{x}_t)]\right) = \frac{1}{\alpha_t}\left(\mathbf{x}_t - \frac{\beta_t}{\sqrt{1-\bar{\alpha}_t}}\boldsymbol{\epsilon}_{\theta^{\mathrm{Q}}}(\mathbf{x}_t, t)\right),
\end{aligned} \tag{15}$$

where $\theta^{\mathrm{Q}}$ and $\theta^{\mathrm{fp}}$ are the noise estimated by the quantized DM and full-preciison counterpart. In Eq. (15), $\boldsymbol{\epsilon}_\theta$ is a function approximator intended to predict $\boldsymbol{\epsilon}$ from $\mathbf{x}_t$. Therefore, Eq. (14) is simplified to:

$$L_{t-1} = \mathbb{E}_{\mathbf{x}_0, \boldsymbol{\epsilon}}\left[\frac{\beta_t^2}{2\tilde{\beta}_t\alpha_t(1-\bar{\alpha}_t)}\|\boldsymbol{\epsilon}_{\theta^{\mathrm{FP}}}(\sqrt{\bar{\alpha}_t}\mathbf{x}_0 + \sqrt{1-\bar{\alpha}_t}\boldsymbol{\epsilon}, t) - \boldsymbol{\epsilon}_{\theta^{\mathrm{Q}}}(\sqrt{\bar{\alpha}_t}\mathbf{x}_0 + \sqrt{1-\bar{\alpha}_t}\boldsymbol{\epsilon}, t)\|^2\right]. \tag{16}$$

With the aforementioned derivation and parameterization, we have the final objective of our noise-estimating imitation, which is formulated as:

$$\begin{aligned}
\arg\min_{\theta^{\mathrm{Q}}} \ & L_{\mathrm{NeM}}(\theta^{\mathrm{Q}}, \theta^{\mathrm{FP}}) \\
& := \mathbb{E}_{t, \mathbf{x}_0, \boldsymbol{\epsilon}}\left[\|\boldsymbol{\epsilon}_{\theta^{\mathrm{FP}}}(\sqrt{\bar{\alpha}_t}\mathbf{x}_0 + \sqrt{1-\bar{\alpha}_t}\boldsymbol{\epsilon}, t) - \boldsymbol{\epsilon}_{\theta^{\mathrm{Q}}}(\sqrt{\bar{\alpha}_t}\mathbf{x}_0 + \sqrt{1-\bar{\alpha}_t}\boldsymbol{\epsilon}, t)\|^2\right],
\end{aligned} \tag{17}$$

## 5 Experiments

In this section, we evaluate the proposed Q-DM framework on several popular diffusion models (*i.e.* DDPM [10] and DDIM [32]) for unconditional image generation. To the best of our knowledge, there is no published work done on low-bit quantized diffusion models at this point, so we report LSQ [5] as a baseline. Experiments show our approach can achieve competitive generation quality to the full-precision scenario on all experimental settings under low-bit quantization.

Table 2: Experiment on 2/3/4-bit quantized diffusion models generating CIFAR-10 [13] image or ImageNet [14] image. "#Bits" denotes the bit-width of weights/activations. "Reso." represents the generating resolution.

| Model | Dataset & Reso. | Step | Method | #Bits | Size$_{(MB)}$ | OPs$_{(G)}$ | FID↓ | IS↑ |
|---|---|---|---|---|---|---|---|---|
| DDIM | CIFAR-10 32×32 | 50 | Full-precision | 32/32 | 4.47 | 390.4 | 4.67 | 9.27 |
| | | | PTQ4DM [6] | 8/8 | 1.12 | 99.5 | 18.02 | 8.87 |
| | | | Baseline | 4/4 | 0.56 | 49.9 | 10.22 | 8.91 |
| | | | **Q-DM** | 4/4 | 0.56 | 49.9 | **6.89** | **8.96** |
| | | | Baseline | 3/3 | 0.28 | 25.1 | 13.24 | 8.88 |
| | | | **Q-DM** | 3/3 | 0.28 | 25.1 | **9.07** | **8.98** |
| | | | Baseline | 2/2 | 0.14 | 12.6 | 18.74 | 8.65 |
| | | | **Q-DM** | 2/2 | 0.14 | 12.6 | **15.26** | **8.86** |
| DDIM | CIFAR-10 32×32 | 100 | Full-precision | 32/32 | 4.47 | 780.7 | 4.16 | 9.32 |
| | | | PTQ4DM [6] | 8/8 | 1.12 | 199.0 | 14.18 | 9.31 |
| | | | Baseline | 4/4 | 0.56 | 99.8 | 9.02 | 8.95 |
| | | | **Q-DM** | 4/4 | 0.56 | 99.8 | **5.12** | **9.21** |
| | | | Baseline | 3/3 | 0.28 | 50.1 | 12.24 | 8.90 |
| | | | **Q-DM** | 3/3 | 0.28 | 50.1 | **8.12** | **8.94** |
| | | | Baseline | 2/2 | 0.14 | 25.2 | 16.99 | 8.74 |
| | | | **Q-DM** | 2/2 | 0.14 | 25.2 | **14.31** | **8.77** |
| DDPM | CIFAR-10 32×32 | 1000 | Full-precision | 32/32 | 4.47 | 7807.2 | 3.17 | 9.46 |
| | | | PTQ4DM [6] | 8/8 | 1.12 | 1990.0 | 7.10 | 9.55 |
| | | | Baseline | 4/4 | 0.56 | 997.7 | 9.11 | 8.96 |
| | | | **Q-DM** | 4/4 | 0.56 | 997.7 | **5.17** | **9.15** |
| | | | Baseline | 3/3 | 0.28 | 501.0 | 12.28 | 8.91 |
| | | | **Q-DM** | 3/3 | 0.28 | 501.0 | **8.14** | **8.93** |
| | | | Baseline | 2/2 | 0.14 | 252.0 | 16.93 | 8.72 |
| | | | **Q-DM** | 2/2 | 0.14 | 252.0 | **14.35** | **8.76** |
| DDIM | ImageNet 64×64 | 50 | Full-precision | 32/32 | 4.47 | 390.4 | 20.57 | 15.72 |
| | | | PTQ4DM [6] | 8/8 | 1.12 | 99.5 | 25.87 | 14.99 |
| | | | Baseline | 4/4 | 0.56 | 49.9 | 24.78 | 15.37 |
| | | | **Q-DM** | 4/4 | 0.56 | 49.9 | **20.02** | **15.68** |
| | | | Baseline | 3/3 | 0.28 | 25.1 | 26.35 | 15.24 |
| | | | **Q-DM** | 3/3 | 0.28 | 25.1 | **22.19** | **15.32** |
| | | | Baseline | 2/2 | 0.14 | 12.6 | 32.43 | 14.66 |
| | | | **Q-DM** | 2/2 | 0.14 | 12.6 | **28.42** | **15.03** |
| DDIM | ImageNet 64×64 | 100 | Full-precision | 32/32 | 4.47 | 780.7 | 19.70 | 15.98 |
| | | | PTQ4DM [6] | 8/8 | 1.12 | 199.0 | 24.92 | 15.52 |
| | | | Baseline | 4/4 | 0.56 | 99.8 | 24.46 | 15.51 |
| | | | **Q-DM** | 4/4 | 0.56 | 99.8 | **19.56** | **15.92** |
| | | | Baseline | 3/3 | 0.28 | 50.1 | 26.23 | 15.42 |
| | | | **Q-DM** | 3/3 | 0.28 | 50.1 | **21.97** | **15.92** |
| | | | Baseline | 2/2 | 0.14 | 25.2 | 31.19 | 14.89 |
| | | | **Q-DM** | 2/2 | 0.14 | 25.2 | **27.94** | **14.99** |
| DDPM | ImageNet 64×64 | 1000 | Full-precision | 32/32 | 4.47 | 7807.2 | 18.98 | 16.63 |
| | | | PTQ4DM [6] | 8/8 | 1.12 | 1990.0 | 22.32 | 15.31 |
| | | | Baseline | 4/4 | 0.56 | 997.7 | 22.91 | 15.29 |
| | | | **Q-DM** | 4/4 | 0.56 | 997.7 | **18.52** | **16.72** |
| | | | Baseline | 3/3 | 0.28 | 501.0 | 24.75 | 15.11 |
| | | | **Q-DM** | 3/3 | 0.28 | 501.0 | **20.21** | **16.17** |
| | | | Baseline | 2/2 | 0.14 | 252.0 | 29.33 | 14.87 |
| | | | **Q-DM** | 2/2 | 0.14 | 252.0 | **25.62** | **15.48** |

## 5.1 Datasets and Implementation Details

We evaluate our method on two datasets including 32×32 generating size in CIFAR-10 [13] and 64×64 generating size in ImageNet [14]. For the CIFAR-10 [13] and ImageNet [14] datasets, we use the DDIM [32] sampler with 50/100 sampling timesteps and DDPM [10] with 1000 sampling timesteps. All the training settings are the same as DDPM [10]. For DDIM sampler, we set $\eta$ in DDIM [32] as $0.5$ for the best performance. We evaluate the performance of our method using FID [9] and Inception Score (IS) [27] on both CIFAR-10 [13] and ImageNet [14] datasets. We set the training timestep $T = 1000$ for all experiments, following [10]. We set the forward process variances to constants increasing linearly from $\beta_1 = 1e - 4$ to $\beta_T = 0.02$. To represent the reverse process, we use a U-Net backbone, following [10, 32]. Parameters are shared across time, which is specified to the network using the Transformer sinusoidal position embedding [36]. We use self-attention at the 16×16 feature map resolution [36, 37].

## 5.2 Ablation Study

We give quantitative results of the proposed TaQ and NeM in Tab. 1 As can be seen, the low-bit quantized DM baseline [5] suffers a severe performance drop on image generation task compared with full-precision DMs (5.55, 8.57, and 14.07 performance gap in terms of FID score with 4/3/2-bit, respectively). TaQ and NeM improve the performance of generation when used alone. For example, the 4-bit quantized DM basline with TaQ and NeM introduced separately achieves 0.97 and 1.24 FID score decrease, respectively.

Moreover, the two techniques further boost the performance considerably when combined together. For instance, when combining the TaQ and NeM together, the performance of 4/3/2-bit quantized DMs improvement achieves 3.33, 4.07, and 3.48 respectively. To conclude, the two techniques can promote each other to improve Q-DM and close the performance gap between low-bit quantized DMs and full-precision counterpart.

## 5.3 Main Results

The experimental results are shown in Tab. 2. We compare our method with 4/3/2-bit baseline [5] based on the same frameworks for the task of unconditional image generation with the CIFAR-10 [13] and ImageNet [14] dataset. We also report the classification performance of the 8-bit PTQ method, *i.e.*, PTQ4DM [30]. We firstly evaluate the proposed method on CIAFR-10 [13] with DDIM [32] and DDPM [10]. We use the model size and OPs (defined in [18]) to evaluate the efficiency of quantized and full-precision models.

For 50-step DDIM sampler, compared with 8-bit PTQ4DM [30], our 4-bit Q-DM achieves a much larger compression ratio than 8-bit PTQ4DM, but with significant performance improvement (6.89 FID ↓ *vs.* 18.02 FID ↓). And it is worth noting that the proposed 2-bit model significantly compresses the DDIM by 30.9× on OPs. The proposed method boosts the performance of 4/3/2-bit Baseline by 3.33, 4.17, and 3.48 in terms of FID score with the same architecture and bit-width, which is significant on the CIFAR-10 [13] dataset with 32×32 generating resolution. For 1000-step DDPM, the performance of the proposed method outperforms the 4/3/2-bit Baseline by 3.94, 4.14, and 2.58, a large margin. Also note that the proposed 4/3/2-bit model significantly accelerates the generation by 7.8×, 15.6×, and 30.9× on OPs. Compared with 8-bit PTQ4DM, ours achieve significantly higher compression and acceleration rate, while the performance improvement is considerable.

Also, our method generates convincing results on ImageNet [14] dataset. As shown in Tab. 2, the performance of the proposed method with 50-step DDIM significantly outperforms the 4/3/2-bit Baseline method by 4.76 , 4.16, and 4.01. Compared with 8-bit PTQ method, our method achieves significantly higher compression rate and acceleration rate, but with better performance. For 1000-step DDPM on ImageNet [14] dataset, the performance of the proposed method outperforms the 4/3/2-bit Baseline by 4.39, 4.54, and 3.71. Also note that our 4-bit Q-DM surpasses the full-precision 50/100-step DDIM and 1000-step DDPM and significantly compresses the noise estimation networks by 7.9× , which demonstrates the effectiveness and efficiency of our Q-DM.

# 6 Conclusion

In this paper, we present Q-DM, an efficient low-bit quantized diffusion model that offers a high compression ratio and competitive performance in image generation task. Initially, we analyze the challenges of the low-bit quantized DM. Our empirical analysis show that distribution oscillation in activation is the one of the cause of the significant drop in DM quantization. Another challenge lies in the accumulated quantization error resulted from the multi-step denoising process during inference. To address these issues, we first develop a timestep-aware quantization (TaQ) method and a noise-estimating mimicking (NeM) scheme for low-bit quantized DMs, to effectively address these two challenges. Our work provides a comprehensive analysis and effective solutions for the crucial issues in low-bit quantized diffusion model, paving the way for the extreme compression and acceleration of diffusion model.

# 7 Acknowledgement

This work was supported in part by the National Natural Science Foundation of China under Grant 62076016, under Grant 61827901, "One Thousand Plan" projects in Jiangxi Province Jxsg2023102268, Foundation of China Energy Project GJNY-19-90, the National Key R&D Program of China (NO.2022ZD0160100).

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
