# OpenReview forum: "Q-DM: An Efficient Low-bit Quantized Diffusion Model"
_NeurIPS.cc/2023/Conference — NeurIPS 2023 poster_

### Official Review · Reviewer_3MiP · 2023-07-04

**Soundness:** 3 good
**Presentation:** 4 excellent
**Contribution:** 3 good
**Rating:** 7
**Confidence:** 4

**Summary:**

This paper presents a quantization-aware training framework for diffusion models. The authors identified activation distribution oscillation and quantization error accumulation as the main causes of the performance drop. To close the performance gap, the authors developed a timestep-aware quantization (TaQ) method for data normalization and a noise-estimating mimicking (NeM) training scheme. Experiments were conducted to demonstrate the effectiveness of the proposed method.

**Strengths:**

1. The motivation of this paper is valid. The proposed method is reasonable. The shifting of the activation distributions and the quantization error accumulation are quite likely to affect the overall performance. The authors propose Timestep-aware Quantization (TaQ) and Noise-estimating Mimicking (NeM) accordingly to address these two problems.
2. The paper is well-organized and the writing is fluent. This helps the readers capture the high-level ideas well and the detailed experimental setups make this paper easy to replicate the results.
3. The experimental results are promising. Q-DMs demonstrate performance on par with FP models.

**Weaknesses:**

1. Will the TaQ quantization process affect the inference speed? See 3.b in the ``Questions'' section.
2. Some technical details should be clarified (see the questions below).

**Questions:**

1. What does "PTQ4DM" exactly refer to in Tab. 2? Tab.2 cites [1] while referring PTQ4DM to [2] in Line 243. As far as I am concerned, PTQ4DM refers to a recently published work [2], presenting a method for calibration data collection in diffusion quantization, then what does Line 242 mean by saying "we also report the **classification performance**..." ?

2. I have some questions regarding the method details, I would like to ask for some clarifications:
    a. Did you calculate the statistical mean and variance of each activation in an offline manner, i.e. the statistical values were pre-computed with the full-precision DMs and not updated with the training procedure?
    b. How did you conduct inference with such the timestep-aware design? Did you conduct activation normalization with the pre-computed values on-the-fly? If so, how would it affect the inference efficiency?
   c. How is the "distance" in Figure 4 calculated? Also, is the activation distribution in Figure 3 layer-wise or channel-wise?
   d. Is it possible to combine the original training loss in DMs with the NeM loss?

[1] Post-training piecewise linear quantization for deep neural networks

[2] Post-Training Quantization on Diffusion Models

**Limitations:**

The authors do not include the limitations and potential negative societal impact in the paper.

---

> ### Author Rebuttal · Authors · 2023-08-09
>
> **Q1**: What does "PTQ4DM" exactly refer to in Tab. 2? Tab.2 cites [1] while referring PTQ4DM to [2] in Line 243. As far as I am concerned, PTQ4DM refers to a recently published work [2], presenting a method for calibration data collection in diffusion quantization, then what does Line 242 mean by saying "we also report the classification performance..." ?
>
> **A1**: Sorry for the typo, the citation should be “Post-training quantization on diffusion models [2]”, and the Line 242 should be “We also report the generation performance of the 8-bit PTQ method”. The typos will be revised it in the final version.
>
> **Q2(a)**:  Did you calculate the statistical mean and variance of each activation in an offline manner, i.e. the statistical values were pre-computed with the full-precision DMs and not updated with the training procedure?
>
> **A2(a)**: Only the statistical values in Fig. 3 are calculated by a full-precision diffusion model in an offline manner, to explain our motivation. In our QAT process, the statistical mean and variance of each activation are calculated on-the-fly. We apology for the typo and misleading in the caption. We will polish it in the final version.
>
> **Q2(b)**: How did you conduct inference with such the timestep-aware design? Did you conduct activation normalization with the pre-computed values on-the-fly? If so, how would it affect the inference efficiency?
>
> **A2(b)**: The calculation of statistical mean and variance is performed on-the-fly during inference. As shown in Tab. 2, the increase of flops is negligible. (Copy from Q2 of R\#bfDz).
>
> **Q2(c)**: How is the "distance" in Figure 4 calculated? Also, is the activation distribution in Figure 3 layer-wise or channel-wise?
>
> **A2(c)**: In Fig. 4, we calculate the distance between the same layer of the full-precision DMs and quantized counterparts. In Fig. 3, the activation distribution is about all elements in one specific layer, i.e., layer-wise distribution. We will add these detailed descriptions in the final version.
>
> **Q2(d)**: Is it possible to combine the original training loss in DMs with the NeM loss?
>
> **A2(d)**: We have conducted experiments of combining the original training loss in DMs with the NeM loss. We evaluate the combination of original loss in DDPM [1] and NeM with 4-bit Q-DM based on 50-step DDIM sampler with 32×32 generating resolution on CIFAR-10, in below. As can be seen, this combination affects the performance of Q-DM. The best result (in **bold**) is achieve by singly combining the 4-bit baseline with our NeM model. Hence, we find that the NeM method performs best when singly being applied. These results and analysis will be added in the final version.
>
>
>
>  | Method | FID$\downarrow$ | IS $\uparrow$|
>  | ------ | ----- | ------ |
>  | Full-precision | 4.67 | 9.27 |
>  | 4-bit Baseline (LSQ) + Original Loss [1] | 10.22 | 8.91 |
>  | **4-bit Baseline (LSQ) + NeM** | **8.98** | **8.92** |
>  | 4-bit Baseline (LSQ) + Original Loss [1] + NeM | 9.05 | 8.87 |
>
> [1] Post-training Piecewise Linear Quantization for Deep Neural Networks. ECCV'2020.
>
> [2] Post-Training Quantization on Diffusion Models. CVPR'2023.
>
> [3] Denoising Diffusion Probabilistic Models. NeurIPS'2020.

---

> > ### Comment · Reviewer_3MiP · 2023-08-17
> > **Thanks for the response**
> >
> > Thanks for the authors' response. My concerns have been addressed. I keep my rating as Accept.

---

> > > ### Author Response · Authors · 2023-08-19
> > > **Thanks for the reviewing**
> > >
> > > We express our appreciation for the time you dedicated to reviewing our paper.

---

### Official Review · Reviewer_bfDz · 2023-07-05

**Soundness:** 2 fair
**Presentation:** 2 fair
**Contribution:** 2 fair
**Rating:** 4
**Confidence:** 5

**Summary:**

The paper proposed a quantization-aware training scheme for diffusion models, based on the well-known method, LSQ.
In the paper, they identified the bottleneck come from a large distribution oscillation on activations and accumulated quantization error caused by the denoising process.
Then, they suggest method to address the issues:
Timestep-aware quantization (TaQ) – Time Step-aware activation smoothing to handle a large oscillation on activation
Noise-estimating Mimicking (NeM):  reduceing error accumulation throughout the multi-step denoising process with quantization


**Strengths:**

They observed and identified the issues when compressing DMs in a quantization-aware training manner.

**Weaknesses:**

1.	The suggested methods to tackle the issues in QAT for DM are not novel. TaQ and NeM seem just normalization and knowledge distillation which can be commonly used in the related literature.
2.	There is much lack of numerical descriptions for the QAT process, such as training time, the number of calibration datasets, and the number of epochs, etc. They have to provide detail about how many resources are needed for reproduction.
3.	The models used in the paper to show their performance are a little bit outdated. The datasets they used are also insufficient to prove the performance. They should provide more thorough experimental results with recent models and datasets.


**Questions:**

1.	Generally, QAT requires more data and training time than PTQ, and U-net inherently has a large number of parameters, so requires a lot of training data and takes a long time to optimize. How much data is used for QAT for DM? And how much time did you take to complete the quantization for U-net?
2.	When performing inference, computing the statistical mean and variance are needed every time step? If so it imposes the additional cost for H/W. How are these values considered at inference time?
3.	As a naive approach, we may have a different step size of activation for each time step to handle the oscillation on activations. Is there any problem with the naive approach compared to your approach?
4.	PTQ4DM and DFQ also handle the dynamic range of activations. Compared with them, what are the advantages of your method?
5.	The problem you observed in quantizing DM may have been dealt with in other literature, even if it is not related to quantization and DM. For example, there are many approaches to normalize the value. The paper would be more robust if the authors provide sufficient related works handling the issues the authors addressed.
6.	In Eq(11) and (12), TaQ quantize normalized activations. Is there no need for compensation for the normalization?


**Limitations:**

See weakness and questions.

---

> ### Author Rebuttal · Authors · 2023-08-09
>
> Due to the character number constraint of rebuttal, we abridge the question in the rebuttal part. All experiments below are conducted on 50-step DDIM sampler with 32×32 generating resolution on CIFAR-10 dataset.
>
> **Q1**: Regarding TaQ and NeM.
>
> **A1**: Our method is proposed based on the observed activation oscillation and quantification error accumulation phenomena. And the QAT method used for the diffusion model is far from being explored. The experimental results also show the advantages of our method.
>
> **Q2**: Regrading numerical descriptions for the QAT process.
>
> **A2**: Sorry for the missing. The training epoch and training time of DDPM and DDIM is 80k step, which needs 6 GPU days. In the QAT of diffusion models, there is no calibration dataset. We will these description in the final version.
>
> **Q3**: Regarding experiments with recent models and datasets.
>
> **A3**: We have conducted experiments of other models and datasets. Please also see the Q3 & Q4 of Reviewer mdg9.
>
> **Q4**: Regarding the data and training cost of Q-DM.
>
> **A4**: We employ the complete set of images from the training dataset, which are same as  other  QAT methods [3-5]. The training cost comparison  between Q-DM and baseline LSQ is shown below. Our Q-DM is about 1.6% lower than baseline method but achieves higher performance. We will add  explanations in the final version for clarity.
>
> |Method|Training time (gpu days)|
> |--|--|
> |Baseline|6.0|
> |Q-DM|6.1|
>
> **Q5**: Regarding computation cost of TaQ.
>
> **A5**: We reply to this question point-to-point.
> - Yes, each step requires computation, but the computational cost is negligible. The OPs value are shown in Tab. 2 of the submission, which include the inference costs of Q-DM  with or without the mean and variance computation.
>
> - We further detailed calculate the OPs of Q-DM and TaQ module, as an example, in the below table. As shown, the extra cost of TaQ module is counted less than 1% in terms of total OPs, which is almost negligible during inference.
>
> |Method|\#Bits|OPs|TaQ OPs (ratio)|
> |--|--|--|--|
> |Full-precision|32/32|390.4|- (-)|
> |Q-DM|4/4|49.9|0.10 (0.2%)|
> |Q-DM|3/3|25.1|0.10 (0.4%)|
> |Q-DM|2/2|12.6|0.10 (0.8%)|
>
> **Q6**: Regarding different step size
>
> **A6**: Using different step sizes for each weight at every time step will result in increased storage and computational burden. As shown below, our method achieves higher compression and acceleration ratio.
>
> |Method|\#Bits|Size|OPs|
> |--|--|--|--|
> |Q-DM |4/4|0.56|49.98|
> |Different step size|4/4|0.57|50.01|
> |Q-DM|3/3|0.28|25.12|
> |Different step size|3/3|0.29|25.15|
> |Q-DM|2/2|0.14|12.66|
> |Different step size|2/2|0.15|12.69|
>
> **Q7**: Regarding advantages of Q-DM.
>
> **A7**: We state the difference and advantages of out method in two aspects: in technology and in performance.
>
> - **Regarding technology**: PTQ4DM [6] is a PTQ method designed for the diffusion model, which can partially address the activation oscillation issue by constructing a more suitable calibration dataset (not applicable in the QAT process, see Fig. 4 in [7]). DFQ [8] eliminates this phenomenon through cross-layer equalization for backbones regarding image classification and detection task, not suitable for the QAT process of DMs. (also a particular phenomenon in the PTQ process), which is not suitable for QAT methods. In contrast, our method aims to improve the QAT process for DMs and successfully eliminates the activation oscillation issue, from a perspective of QAT-specific technology.
>
> - **Regarding performance**: Compared with PTQ methods, the proposed Q-DM also benefits from the main advantage of the QAT process, i.e., superior performance with lower bit-width precision. We further evaluate the PTQ4DM and DFQ in 8- and 4-bit bit-width, where the results are shown in below. We have two main observations: 1) the 4-bit Q-DM achieves better performance  that 8-bit PTQ methods while possessing lower-precision weights and activations; 2) the PTQ methods deteriorates in 4-bit format, while Q-DM performs well with lower precision. These observations demonstrate the superiority of the Q-DM method.
>
>  |Method|\#Bits|FID$\downarrow$|IS$\uparrow$|
>  |--|--|--|--|
>  |Full-precision|32/32|4.67|9.27|
>  |PTQ4DM |8/8|18.02|8.87|
>  |DFQ|8/8|18.96|8.83|
>  |PTQ4DM|4/4|19.78|8.76|
>  |DFQ |4/4|20.02|8.68|
>  |**Q-DM**|4/4|**8.98**|**8.92**|
>
> **Q8**: Difference of Q-DM and other normalization.
>
> **A8**: Different from prior works, our TaQ  numerically analyzes the activation ranges across different timesteps and effectively mitigates distribution oscillation specifically for low-bit quantized DMs. To validate, we show some experimental comparisons with the prior methods which normalize the activation value as below. We will add more references and this comparison in the final version for clarity.
>
>  |Method|\#Bits|FID$\downarrow$|IS$\uparrow$|
>  |--|--|--|---|
>  |Q-ViT|4/4|9.48|8.85|
>  |**Q-DM**|4/4|**8.98**|**8.92**|
>
> **Q9**: Regarding compensation for normalization?
>
> **A9**: We conduct experiments regarding normalization and its compensation as below. As shown, our TaQ achieves better performance with less extra parameters introduced. This will be added in the final version.
>
>  |Formulation of Eq. (11)|FID$\downarrow$|IS$\uparrow$|
>  |--|--|--|
>  |**a** (Baseline) |8.98|8.92|
>  |**Norm(**a**) (TaQ)**|**6.89**|**8.96**|
>  |Norm(**a**)*scale |6.92|8.95|
>  |Norm(**a**)*scale+bias|6.91|8.95|
>
> [1] Denoising Diffusion Probabilistic Models. NeurIPS'2020.
>
> [2] Denoising Diffusion Implicit Models. ICLR'2021.
>
> [3] Q-ViT: Accurate and Fully Quantized Low-bit Vision Transformer. NeurIPS'2022.
>
> [4] Q-DETR: An Efficient Low-Bit Quantized Detection Transformer. CVPR’2023.
>
> [5] Post-Training Quantization on Diffusion Models. CVPR'2023.
>
> [6] A Survey of Quantization Methods for Efficient Neural Network Inference. ArXiv: 2103.13630.
>
> [7] Data-Free Quantization Through Weight Equalization and Bias Correction. ICCV'2019.

---

> > ### Comment · Reviewer_bfDz · 2023-08-18
> > **Thank you**
> >
> > Thank you for your effort to address my concern. I acknowledge your work applying QAT to diffusion models for the first time (which might be helpful when quantizing the models), but I am still concerned about the originality and novelty of the paper. so I keep the score as it stands.

---

> > > ### Author Response · Authors · 2023-08-19
> > > **Thanks for the reviewing**
> > >
> > > We extend our gratitude for your diligent review of our paper. It is important to underscore that we introduce QAT technology into diffusion models for the first time, yielding notable outcomes in quantized diffusion models with 2/3/4 bit-widths.

---

### Official Review · Reviewer_6Qdf · 2023-07-06

**Soundness:** 3 good
**Presentation:** 3 good
**Contribution:** 4 excellent
**Rating:** 7
**Confidence:** 5

**Summary:**

In this paper, a novel method called Q-DM is introduced, which enables the creation of low-bit quantized diffusion models. The authors first give extensive analysis about two primary challenges faced by low-bit quantized DMs: significant distribution oscillation on activations and accumulated quantization error arising from the multi-step denoising process. To address these issues, the authors propose two techniques: Timestep-aware Quantization (TaQ) and Noise-estimating Mimicking (NeM). TaQ is designed to mitigate the distribution oscillation problem, while NeM aims to reduce the accumulated quantization error. By incorporating these techniques into Q-DM, the paper demonstrates its ability to overcome these challenges effectively, leading to superior performance compared to existing methods. The experimental results provided in the paper serve as evidence of the high-quality performance achieved by Q-DM.

**Strengths:**

1.This paper is easy to follow. The organization of this paper are exemplary, as it effectively presents the proposed Q-DM in a clear and comprehensible manner. The paper provides comprehensive explanations of how Q-DM improved the performance of quantized DMs.

2.This paper presents an novel quantization method known as Q-DM, which addresses several challenges in low-bit quantized diffusion models. The authors propose the Timestep-aware Quantization (TaQ) method to tackle the issue of activation distribution oscillation, which is caused by the random-sampled timestep during training. Additionally, they introduce the Noise-estimating Mimicking (NeM) scheme to effectively minimize accumulated errors.

3.The experimental results are significant. The Q-DMs exhibit performance comparable to that of full-precision models, showcasing the effectiveness of the proposed methods.



**Weaknesses:**

1.Is the model quantized during all the training and sampling process? Since these may lead to different inference speed. And are the quantization-related parameter all the same across different timestep?

2.Is the proposed TaQ method only used in Q-AttnBlock module? And how is the conv layer quantized in Q-DM?

3.In Table 2, how is the ‘OPs’ calculated?  It would be more detailed if authors can provide some description about this metric.


**Questions:**

See weaknesses.

**Limitations:**

Yes

---

> ### Author Rebuttal · Authors · 2023-08-09
>
> **Q1**: Is the model quantized during all the training and sampling process? Since these may lead to different inference speed. And are the quantization-related parameter all the same across different timestep?
>
> **A1**: Yes, the model is quantized during both the training (to quantized value in floating-point format [1]) and sampling process. Also, the quantization-related parameter are all the same across different timestep during inference.
>
> **Q2**: Is the proposed TaQ method only used in Q-AttnBlock module? And how is the conv layer quantized in Q-DM?
>
> **A2**: The proposed TaQ method is only deployed in the Q-AttnBlock module (Eq. (12)). The quantization method of convolution layer in Q-DM is same as LSQ [2], i.e., the baseline method.
>
> **Q3**: In Table 2, how is the ‘OPs’ calculated? It would be more detailed if authors can provide some description about this metric.
>
> **A3**: The OPs is calculated through "the respective number of FLOPs adds {$\frac {1}{32}$, $\frac {1}{16}$, $\frac {1}{8}$} of the number of {$2$, $3$, $4$}-bit multiplications equals the OPs" following [3].  We will add detailed description in the final version.
>
> [1] A Survey of Quantization Methods for Efficient Neural Network Inference. ArXiv: 2103.13630.
>
> [2] Learned Step Size Quantization. ICLR'2020.
>
> [3] Q-DETR: An Efficient Low-Bit Quantized Detection Transformer. CVPR'2023.

---

### Official Review · Reviewer_acST · 2023-07-06

**Soundness:** 2 fair
**Presentation:** 2 fair
**Contribution:** 2 fair
**Rating:** 5
**Confidence:** 4

**Summary:**

The paper proposes two method to mitigate the accuracy degradation caused by quantization of diffusion models: one is time-step aware quantization (different calibration data and range for each time-step of diffusion), the other is using a full-precision network for training time distillation. Experiments on image generation in Cifar-10 and ImageNet datasets show the efficacy of the method.

**Strengths:**

By using QAT, the proposed method manages to bring down the bitwidth of the neural network to lower than 8-bit. The FID and IS metrics support the efficacy of the method.

**Weaknesses:**

The benefits of per-time-step quantization has been known since earlier works like PTQ4DM and q-diffusion. It is desirable to know the proposed method and prior works. Note that QAT and PTQ do not make much difference here, as the underlying motivation of per-time-step quantization to deal with changing dynamic range change is the same here. Also having per-time-step quantization is almost zero-overhead as the weights need be loaded for computation of each step anyway.

Also having a full-precision network as teacher for knowledge distillation (including having full precision or higher precision teacher) is quite a standard approach. It's not obvious what specialties of QAT are incorporated here.

q-diffusion: https://arxiv.org/abs/2302.04304

**Questions:**

Can visual results be provided for intuitive inspection? In particular, spotty noise may not be well captured by FID/IS metrics but will be blatant when doing visual inspection.

In Table 2, there are a few pairs of results with similar IS but very different FID, like PTQ4DM CIFAR-10 32×32 50 steps and Baseline CIFAR-10 32×32 3/3 . Any discussions of this variation?

**Limitations:**

No particular limitations.

---

> ### Author Rebuttal · Authors · 2023-08-09
>
> **Q1**: The benefits of per-time-step quantization has been known since earlier works like PTQ4DM [1] and Q-Diffusion [2]. It is desirable to know the proposed method and prior works. Note that QAT and PTQ do not make much difference here, as the underlying motivation of per-time-step quantization to deal with changing dynamic range change is the same here. Also having per-time-step quantization is almost zero-overhead as the weights need be loaded for computation of each step anyway.
>
> **A1**: QAT is technically different from PTQ, and also QAT can achieve a better performance than PTQ as shown in our experimental parts. Previous PTQ methods for quantized DM, e.g., PTQ4DM [1], partially address the activation oscillation issue by constructing a calibration dataset, which is not applicable in the QAT process (see Fig. 4 in [3]). Differently, our method addressed this problem by introducing TaQ and NeM into the QAT framework, which can well reduce the activations oscillation and effectively improve the performance of Q-DM. We will add more related work to clarify the difference from our method in the final version.
>
> **Q2**: Also having a full-precision network as teacher for knowledge distillation (including having full precision or higher precision teacher) is quite a standard approach. It's not obvious what specialties of QAT are incorporated here.
>
> **A2**: Although using knowledge distillation is a standard approach in network quantization, the KD method for quantized DMs remains largely under-developed. As one of our contributions, we propose a new knowledge distillation method, called Noise-estimating Mimicking (NeM), to enhance the performance of quantized DMs. We provide the motivation and theoretical derivation of NeM in Sec. 4.2. As shown in Tab.1, the proposed NeM significantly improves the performance of the baseline method, which validates the motivation and theoretical derivation.
>
> **Q3**: Can visual results be provided for intuitive inspection? In particular, spotty noise may not be well captured by FID/IS metrics but will be blatant when doing visual inspection.
>
> **A3**: Thanks for the advice, we will provide qualitative results in the final version.
>
> **Q4**: In Table 2, there are a few pairs of results with similar IS but very different FID, like PTQ4DM CIFAR-10 32×32 50 steps and Baseline CIFAR-10 32×32 3/3 . Any discussions of this variation?
>
> **A4**: Generally, the IS metric has similar results as shown in our practice and also other methods [1,4], even they are very different on FID. For example, similar phenomenon can also be observed in PTQ4DM [1] (Table 4) and DDPM [4] (Table 1). We will add this discussion in the final version.
>
> [1] Post-Training Quantization on Diffusion Models. CVPR'2023.
>
> [2] Q-Diffusion: Quantizing Diffusion Models. ArXiv:2302.04304.
>
> [3] A Survey of Quantization Methods for Efficient Neural Network Inference. ArXiv: 2103.13630.
>
> [4] Denoising Diffusion Probabilistic Models. NeurIPS'2020.

---

### Official Review · Reviewer_mdg9 · 2023-07-14

**Soundness:** 2 fair
**Presentation:** 2 fair
**Contribution:** 2 fair
**Rating:** 4
**Confidence:** 4

**Summary:**

The paper identifies two challenges in low-bit diffusion models (DMs): activation distribution oscillation and quantization error accumulation. To tackle these challenges, the paper introduces two novel techniques: Timestep-aware Quantization (TaQ) and Noise-estimating Mimicking (NeM). Experimental results demonstrate superior performance compared to previous approaches on CIFAR-10 and ImageNet 64x64 datasets.

**Strengths:**

The paper is easy to follow and low-bit diffusion model is an interesting idea worth to try. The presented ablation studies demonstrate that the proposed methods have a positive impact on quantized diffusion models.

**Weaknesses:**

1. The main contribution of this paper involves the utilization of statistical mean and variance to address distribution oscillation. Essentially, this approach is akin to applying a shift and scale operation following quantized operations, which is a common technique employed in low-bit quantization to achieve balanced quantization bins. Similar methods for weight and activation balancing can be found in [1] and [2, 3] respectively.

2. In Equation (12), the application of TaQ to the softmax attention scores appears questionable. Unlike other activations that typically conform to a Gaussian or bell-shaped distribution, softmax attention scores often exhibit a long-tailed distribution, where the sum of the probabilities is 1. Normalizing the softmax attention scores may disrupt the probabilistic interpretation of these scores, which raises concerns about its appropriateness.

3. The proposed Noise-estimating Mimicking technique necessitates fine-tuning the entire model, which may pose practical challenges for large diffusion models trained on extensive datasets like LAION-5B. Efficient fine-tuning method should be considered.

4. The experiments in the paper are limited to CIFAR-10 and ImageNet 64x64 datasets, with a maximum model size of 4.47M. To provide a comprehensive evaluation, it is recommended to conduct additional experiments on larger models, such as LDM[4], and include commonly used datasets like LSUN and ImageNet 256x256.

5. The accuracy comparison experiment in this paper seems to be inadequate, and there are issues with the citations for the compared method PTQ4DM [5]. Moreover, discrepancies exist between the FID and IS results for full-precision (FP) models in this paper and the results reported in PTQ4DM, which could potentially introduce unfairness in the comparisons. As an example, when employing the DDIM sampler for 100 steps on CIFAR-10, the FID for the FP model is reported as 10.05 in PTQ4DM, whereas this paper indicates a result of 4.16.

[1] Qin, Haotong, et al. "Forward and backward information retention for accurate binary neural networks." CVPR 2020.
[2] Liu, Zechun, et al. "Reactnet: Towards precise binary neural network with generalized activation functions." ECCV 2020.
[3] Wei, Xiuying, et al. "Outlier Suppression+: Accurate quantization of large language models by equivalent and optimal shifting and scaling." arXiv:2304.09145 (2023).
[4] Rombach, Robin, et al. "High-resolution image synthesis with latent diffusion models." CVPR 2022.
[5] Shang, Yuzhang, et al. "Post-training quantization on diffusion models." CVPR 2023.

**Questions:**

How are the model size and OPs calculated? The quantized model's size reported in the paper is precisely 1/bits of the full precision model. However, this is unusual because there must be some parameters that remain in full-precision.

**Limitations:**

I did not identify any issues related to limitations and potential negative societal impact.

---

> ### Author Rebuttal · Authors · 2023-08-09
>
> **Q1**: The main contribution of this paper involves the utilization of statistical mean and variance to address distribution oscillation. Essentially, this approach is akin to applying a shift and scale operation following quantized operations, which is a common technique employed in low-bit quantization to achieve balanced quantization bins. Similar methods for weight and activation balancing can be found in [1] and [2, 3] respectively.
>
> **A1**: To the best of our knowledge, our Q-DM is the first QAT-based work to effectively quantize diffusion models. Our method can well address the activation oscillation problem by introducing Timestep-aware Quantization (TaQ) and Noise-estimating Mimicking (NeM) into the QAT framework, which effectively improve the performance of quantizing DM. Different from quantization of other deep models, our Q-DM involves both training and inference process, in this situation our method is very efficient based on a simple method with the shift and scale operations. You can also refer to Q8 of Reviewer bfDz, where we also compare the performance of Q-DM with other normalization methods.
>
> **Q2**: In Eq. (12), the application of TaQ to the softmax attention scores appears questionable. Unlike other activations that typically conform to a Gaussian or bell-shaped distribution, softmax attention scores often exhibit a long-tailed distribution, where the sum of the probabilities is 1. Normalizing the softmax attention scores may disrupt the probabilistic interpretation of these scores, which raises concerns about its appropriateness.
>
> **A2**: We agree with your point on the Gaussian or bell-shaped distribution,  our proposed TaQ method is exactly applied to $a_q$ , $a_k$  before the softmax operation, rather than directly normalizing the softmax attention score. Details are shown in the line 2 of  Eq. (12). Therefore, Eq. (12) is appropriate and does not disrupt the distribution of attention score. We will visualize the effect of TaQ method on probabilistic interpretation of Q-AttnBlock in the final version for clarity.
>
> **Q3**: The proposed Noise-estimating Mimicking technique necessitates fine-tuning the entire model, which may pose practical challenges for large diffusion models trained on extensive datasets like LAION-5B. Efficient fine-tuning method should be considered.
>
> **A3**: We will add more fine-tuning method experiments in the final version. As shown in the table below, our method also effects on  large diffusion models fine-tuned by LoRA-R8 on extensive datasets like LAION-5B. Our method consistently boosts the performance of baseline method in 2/3/4 bit-width format. We will add the experimental results in the final version.
>
> | Model | Method |\#Bits| FID$\downarrow$ |
> | ------ | ------ | ------ | ------ |
> | LDM-4 | Full-precision | 32/32 |20.68|
> | LDM-4 | Baseline (LSQ) | 4/4 |23.42|
> | LDM-4 | Q-DM | 4/4 | 21.56 |
> | LDM-4 | Baseline (LSQ) | 3/3 | 25.81 |
> | LDM-4 | Q-DM | 3/3 | 23.72 |
> | LDM-4 | Baseline (LSQ) | 2/2 | 27.49 |
> | LDM-4 | Q-DM | 2/2 | 25.96 |
>
> **Q4**: The experiments in the paper are limited to CIFAR-10 and ImageNet 64x64 datasets, with a maximum model size of 4.47M. To provide a comprehensive evaluation, it is recommended to conduct additional experiments on larger models, such as LDM [4], and include commonly used datasets like LSUN and ImageNet 256x256.
>
> **A4**: We apply our Q-DM on LDM [4] and test it on the LSUN dataset, and the results are shown in the table below. As can be seen, the proposed Q-DM also show its superiority on the LDM with larger LSUN dataset. For example, Q-DM boosts the baseline models by about 0.3\%~0.5\% FID score with the same bit-width, which is significant. We will add these experiments in the final version.
>
> | Model | Method |\#Bits| FID$\downarrow$ |
> | ------ | ------ | ------ | ------ |
> | LDM-4 | Full-precision | 32/32 | 2.98 |
> | LDM-4 | Q-Diffusion |8/8 | 3.63 |
> | LDM-4 | Baseline (LSQ) | 4/4 | 3.54 |
> | LDM-4 | Q-DM | 4/4 | 3.01 |
> | LDM-4 | Baseline (LSQ) | 3/3 | 3.68 |
> | LDM-4 | Q-DM | 3/3 | 3.37 |
> | LDM-4 | Baseline (LSQ) | 2/2 | 3.99 |
> | LDM-4 | Q-DM | 2/2 | 3.76 |
>
> **Q5**: The accuracy comparison experiment in this paper seems to be inadequate, and there are issues with the citations for the compared method PTQ4DM [5]. Moreover, discrepancies exist between the FID and IS results for full-precision (FP) models in this paper and the results reported in PTQ4DM, which could potentially introduce unfairness in the comparisons. As an example, when employing the DDIM sampler for 100 steps on CIFAR-10, the FID for the FP model is reported as 10.05 in PTQ4DM, whereas this paper indicates a result of 4.16.
>
> **A5**: We follow the same experiments setting as DDPM [6] and DDIM [7]. The FID score of DDIM sampler for 100 steps on CIFAR-10 is reported as 4.16 in the original paper (Table 1, Page 7 in DDIM [7]), which is the same as our re-implementation.
>
> [1] Forward and Backward Information Retention for Accurate Binary Neural Networks. CVPR'2020.
>
> [2] ReActNet: Towards Precise Binary Neural Network with Generalized Activation Functions. ECCV'2020.
>
> [3] Outlier Suppression+: Accurate quantization of large language models by equivalent and optimal shifting and scaling. ArXiv:2304.09145.
>
> [4] High-Resolution Image Synthesis with Latent Diffusion Models. CVPR'2022.
>
> [5] Post-Training Quantization on Diffusion Models. CVPR'2023.
>
> [6] Denoising Diffusion Probabilistic Models. NeurIPS'2020.
>
> [7] Denoising Diffusion Implicit Models. ICLR'2021.

---

> > ### Comment · Reviewer_mdg9 · 2023-08-17
> > **Follow-up Feedback**
> >
> > Thank you for your response. However, I still have some questions:
> >
> > 1)	Could you clarify the statement "Different from the quantization of other deep models, our Q-DM involves both the training and inference processes"? All methods outlined in [A-C] and the normalization layers are engaged in both the training and inference phases. Additionally, what is the key difference between the shift and scale operations you proposed and the techniques presented in the aforementioned methods?
> > 2)	Currently, even the most advanced QAT methods [D, E] experience significant accuracy loss when confronted with low bit-width scenarios (e.g., 2-bit). **None of the QAT methods in 2-bit quantization** manage to rival the performance of the 8-bit PTQ quantization. Considering that classification serves as a foundational task in computer vision, I wonder why your 2-bit Q-DM model's performance closely approaches that of the 8-bit Q-Diffusion (3.76 versus 3.63 on the LSUN dataset), a model that relies on an advanced reconstruction-based PTQ approach. Are there any undisclosed strategies that might have contributed to this result? If your method indeed excels in classification tasks under 2-bit quantization, its potential impact on the community would be substantial.
> > 3)	Since your method is QAT-based, the implementation details for training DMs are missing. For instance, how many epochs did you train? What optimizer and learning rate did you use? How many computing resources are required to train Q-DM?
> > 4)	For the experimental results in A3, did you evaluate LDM-4 on LAION-5B? How did you measure FID on LAION-5B? Did you use any pre-trained model?
> > 5)	In your rebuttal (to me and to other reviewers), a majority of the experimental results lack specific dataset details. For instance, there are several different datasets in LSUN, like bedrooms and churches.
> >
> > [A] Forward and Backward Information Retention for Accurate Binary Neural Networks. CVPR 2020.
> >
> > [B] ReActNet: Towards Precise Binary Neural Network with Generalized Activation Functions. ECCV 2020.
> >
> > [C] Outlier Suppression+: Accurate quantization of large language models by equivalent and optimal shifting and scaling. ArXiv:2304.09145.
> >
> > [D] Overcoming oscillations in quantization-aware training." International Conference on Machine Learning. PMLR 2022.
> >
> > [E] Q-vit: Accurate and fully quantized low-bit vision transformer." NeurIPS 2022.

---

> > > ### Author Response · Authors · 2023-08-19
> > > **Response to Follow-up Feedback**
> > >
> > > We extend our heartfelt gratitude for your dedication in reviewing our paper.
> > >
> > > We abridge the question into several parts and response point-to-point.
> > >
> > > **Q.I**: Could you clarify the statement "Different from the quantization of other deep models, our Q-DM involves both the training and inference processes"? All methods outlined in [A-C] and the normalization layers are engaged in both the training and inference phases. Additionally, what is the key difference between the shift and scale operations you proposed and the techniques presented in the aforementioned methods?
> > >
> > > **A.I**: Sorry for the confusion. We will  change the statement as:"Our approach specifically addresses the issue of Activation Distribution Oscillation, which is observed in the Q-Attention Block  and caused by diverse sampling timesteps of DMs. " These methods [A-C] perform on each layer of CNNs or language models with more additional parameters. Differently, our approach is  targeted  at  rectifying activations within the Q-Attention Block, by adding on different positions (q and k) from [A-C] with fewer learnable parameters and different dimensions. We implement methods  [B, C] into Quantized DMs by 50-step DDIM sampler with 32×32 generating resolution on CIFAR-10 dataset below. Note that the shift and scale operations in IR-Net [A] can not be used in the low-bit situation. We will add these comparison in the final version.
> > >
> > >  | Method | #Bits | FID$\downarrow$ | IS$\uparrow$ |
> > > | -- | -- | -- | -- |
> > >  | tech. in ReActNet | 4/4 | 9.25 | 8.87 |
> > >  | tech. in Ourlier Suppression++ | 4/4 | 9.67 | 8.82 |
> > >  | Q-DM	| 4/4 | 8.98 | 8.92 |
> > >
> > > **Q.II**: Currently, even the most advanced QAT methods [D, E] experience significant accuracy loss when confronted with low bit-width scenarios (e.g., 2-bit). None of the QAT methods in 2-bit quantization manage to rival the performance of the 8-bit PTQ quantization. Considering that classification serves as a foundational task in computer vision, I wonder why your 2-bit Q-DM model's performance closely approaches that of the 8-bit Q-Diffusion (3.76 versus 3.63 on the LSUN dataset), a model that relies on an advanced reconstruction-based PTQ approach. Are there any undisclosed strategies that might have contributed to this result? If your method indeed excels in classification tasks under 2-bit quantization, its potential impact on the community would be substantial.
> > >
> > > **A.II**: For experiments on LSUN-Bedrooms, we do not use any undisclosed strategies. The performance gap on bit-widths differs   for different tasks, particularly for the image generation task, the gap seems to be smaller than others. We will add more detailed training and evaluating settings in the final version.
> > >
> > > **Q.III**: Since your method is QAT-based, the implementation details for training DMs are missing. For instance, how many epochs did you train? What optimizer and learning rate did you use? How many computing resources are required to train Q-DM?
> > >
> > > **A.III**: Sorry for the missing. On CIFAR-10 dataset, the training epoch and training time of DDPM and DDIM is 80k step, which needs 6 GPU days. More details can be referred to A4 of Reviewer bfDz. We will add these description and other necessary training settings in the final version.
> > >
> > > **Q.IV**: For the experimental results in A3, did you evaluate LDM-4 on LAION-5B? How did you measure FID on LAION-5B? Did you use any pre-trained model?
> > >
> > > **A.IV**: Due to the limitation of timeline, we randomly select 10k image-text pairs from LAION-5B for training and evaluating. The LDM-4 in A3 is evaluated on the selected data. We use a full-precision LDM-4 trained on the selected data as a pre-trained model. We will conduct experiments on the whole LAION-5B dataset and add necessary descriptions in the final version.
> > >
> > > **Q.V**: In your rebuttal (to me and to other reviewers), a majority of the experimental results lack specific dataset details. For instance, there are several different datasets in LSUN, like bedrooms and churches.
> > >
> > > **A.V**: Sorry for the missing. The experiments are conducted on LSUN-Bedrooms with a generation resolution of 256 x 256.

---

> > > > ### Comment · Reviewer_mdg9 · 2023-08-20
> > > > **Thanks for the response**
> > > >
> > > > Thanks for the prompt response from the authors. It partially addressed my concerns and I am willing to raise my rating to 4 as some still exist:
> > > >
> > > > 1: The results are not very convincing especially for the 2-bit variant. I speculate that this might be related to the training and evaluation setting. I am not sure whether the comparison is fair enough due to substantial lack of training details, compounded by the absence of supplementary materials (such as example code) necessary for reproducibility. Additionally, I am intrigued by the duration required to train a sizable LDM model on a big dataset and its associated GPU memory consumption.
> > > >
> > > > 2: Although the approach shows some distinctions from previous works, the overall technical novelty is not that strong which is also pointed out by another Reviewer bfDz.

---

> > > > > ### Author Response · Authors · 2023-08-21
> > > > > **Response to Reviewer mdg9**
> > > > >
> > > > > We thank you for your time in reviewing our paper.
> > > > >
> > > > > **Q1**: The results are not very convincing especially for the 2-bit variant. I speculate that this might be related to the training and evaluation setting. I am not sure whether the comparison is fair enough due to substantial lack of training details, compounded by the absence of supplementary materials (such as example code) necessary for reproducibility. Additionally, I am intrigued by the duration required to train a sizable LDM model on a big dataset and its associated GPU memory consumption.
> > > > >
> > > > > **A1**: To convince, we will release the source code after the acceptance, which make sure that our Q-DM can totally be re-implemented. The training settings of our Q-DM on CIFAR-10 and ImageNet datasets are same as the full-precision models, following the standard setting as in DDPM [1] and DDIM [2], which are described in our submission manuscript and the rebuttal content. We train LDM in the rebuttal phase using 8 NVIDIA A100 gpus for 6 days.
> > > > >
> > > > > **Q2**: Although the approach shows some distinctions from previous works, the overall technical novelty is not that strong which is also pointed out by another Reviewer bfDz.
> > > > >
> > > > > **A2**: We would like to conclude that our method contains a normalization operation to dismiss the activation oscillation phenomenon resulted from the multi-step denoising process in diffusion models. Our method also contains a novel noise-estimating distillation, which is proposed by ourselves for the first time and successfully eliminate the accumulated quantization error. The experimental and ablative results also show the effectiveness of these two methods.

---

### Decision · Program_Chairs · 2023-09-21

**Decision:**

Accept (poster)

**Comment:**

This paper introduces a novel method called Q-DM (Quantized Diffusion Models) to address the challenges faced by low-bit quantized diffusion models. The authors identify activation distribution oscillation and quantization error accumulation as primary concerns and propose two techniques, Timestep-aware Quantization (TaQ) and Noise-estimating Mimicking (NeM), to mitigate these issues. Experimental results on CIFAR-10 and ImageNet datasets validate the superior performance of Q-DM compared to previous approaches.

Reviewers raised concerns about the lack of novelty in the proposed techniques, the need for more detailed numerical descriptions and broader experimental evaluation, and discrepancies in the accuracy comparison and citations. The authors provided a thorough response addressing the concerns raised by the reviewers, leading to a final score of 4, 4, 5, 7, 7. As two reviewers still had doubts about the novelty of the paper, the AC carefully reviewed the paper and all the reviewers' comments.

Considering that this paper is the first to introduce the QAT technique to diffusion models, the AC recognizes its significant value in the quantization of diffusion models. With an average score of 5.4, the paper is competitive for this year's NeurIPS conference. The AC agrees to accept the paper and strongly recommends incorporating the content from the rebuttal into the final version.